# Cost-effectiveness of a domestic violence and abuse training and support programme in primary care in the real world: updated modelling based on an MRC phase IV observational pragmatic implementation study

Estela Capelas Barbosa,[1] Talitha Irene Verhoef,[1] Steve Morris,[1] Francesca Solmi,[2] Medina Johnson,[3] Alex Sohal,[4] Farah El-Shogri,[4] Susanna Dowrick,[4] Clare Ronalds,[5] Chris Griffiths,[4,6] Sandra Eldridge,[4] Natalia V Lewis,[4] Angela Devine,[7,8] Anne Spencer,[9] Gene Feder[10]

For numbered affiliations see end of article.

**Correspondence to**
Dr Estela Capelas Barbosa;
e.barbosa@ucl.ac.uk

## ABSTRACT

**Objectives** To evaluate the cost-effectiveness of the implementation of the Identification and Referral to Improve Safety (IRIS) programme using up-to-date real-world information on costs and effectiveness from routine clinical practice. A Markov model was constructed to estimate mean costs and quality-adjusted life-years (QALYs) of IRIS versus usual care per woman registered at a general practice from a societal and health service perspective with a 10-year time horizon.

**Design and setting** Cost–utility analysis in UK general practices, including data from six sites which have been running IRIS for at least 2 years across England.

**Participants** Based on the Markov model, which uses health states to represent possible outcomes of the intervention, we stipulated a hypothetical cohort of 10 000 women aged 16 years or older.

**Interventions** The IRIS trial was a randomised controlled trial that tested the effectiveness of a primary care training and support intervention to improve the response to women experiencing domestic violence and abuse, and found it to be cost-effective. As a result, the IRIS programme has been implemented across the UK, generating data on costs and effectiveness outside a trial context.

**Results** The IRIS programme saved £14 per woman aged 16 years or older registered in general practice (95% uncertainty interval −£151 to £37) and produced QALY gains of 0.001 per woman (95% uncertainty interval −0.005 to 0.006). The incremental net monetary benefit was positive both from a societal and National Health Service perspective (£42 and £22, respectively) and the IRIS programme was cost-effective in 61% of simulations using real-life data when the cost-effectiveness threshold was £20 000 per QALY gained as advised by National Institute for Health and Care Excellence.

**Conclusion** The IRIS programme is likely to be cost-effective and cost-saving from a societal perspective in the UK and cost-effective from a health service perspective,

## Strengths and limitations of this study

- ► We have used up-to-date routine data from several sites across England to evaluate the value for money of Identification and Referral to Improve Safety (IRIS), a domestic violence training programme.
- ► We were unable to include any impact of the IRIS programme on children exposed to domestic violence and abuse (DVA), as to our knowledge, there are no available cohort studies focusing on the cost and benefits of DVA interventions for this population.
- ► We have used mainly data on short-term outcomes, although modelled long-term outcomes, as to our knowledge, no study has tracked women subject to DVA over long periods of time.

although there is considerable uncertainty surrounding these results, reflected in the large uncertainty intervals.

## INTRODUCTION

The lifetime prevalence of domestic violence and abuse (DVA) against women, including any form of controlling, coercive, threatening behaviour, violence and abuse, as well as non-physical forms of abuse as defined by the United Nations,[1] varies internationally from 15% to 71%.[2] In the UK, in the year ending March 2017, 7.5% of women (1.2 million) experienced domestic abuse.[3] Women who experience DVA suffer chronic health problems including gynaecological problems, gastrointestinal disorders, neurological symptoms, chronic pain, cardiovascular conditions and mental health problems.[4–7] In 2012, the cost of DVA in the UK, including medical

and social services, lost economic output and emotional costs, was estimated to be £11 billion.[8] While such estimates highlight the importance of DVA as a public health and clinical problem, information on cost-effectiveness is needed to make an economic case for investment in DVA interventions in healthcare, particularly when health systems are dominated by austerity.

The Identification and Referral to Improve Safety (IRIS)[9] trial tested the effectiveness of a training and support intervention for general practice teams in two English cities.[10] Discussions about DVA between clinicians and patients were 22 times greater in the intervention practices compared with the control practices. Primary care practices that delivered the intervention also experienced a sixfold and threefold increase in referrals received by DVA agencies and DVA-related notes in the patient medical records, respectively. The IRIS programme can now be commissioned across the UK: as of December 2016, 34 UK areas had commissioned IRIS, >800 general practitioner (GP) practices nationally have had IRIS training and over 5000 women have been referred to DVA support services by IRIS since 2010.

The cost-effectiveness of the IRIS trial was assessed using data from the trial and the programme was estimated to be good value for money.[11] Given its national implementation, IRIS became a real-life, long-term intervention, raising the need for a new economic evaluation outside the trial context. The aim of this study was to evaluate the cost-effectiveness of the IRIS programme now that it has been implemented across the UK. Our estimates use up-to-date figures from an Medical Research Council (MRC) phase IV observational pragmatic implementation study[12] on costs and effectiveness from routine clinical practice and the most up-to-date model input parameters, including a recently updated Cochrane review of domestic violence advocacy.[13]

## METHODS

### Overview of economic evaluation

This was a cost-utility analysis, comparing IRIS with usual care in general practices. The outcome measure was quality-adjusted life years (QALYs), as recommended for economic evaluations in the UK.[14] The main analysis was from a societal perspective, as many of the costs of DVA are borne outside the health system; we also estimated cost utility from a National Health Service (NHS) perspective. Costs were calculated in 2015/2016 UK£. We calculated costs and benefits over a 10-year time horizon, with future costs and outcomes discounted at an annual rate of 3.5%.[14]

### Model structure

We developed a Markov model (figure 1) based on the previous analysis.[11] The model has five states and the cycle length was 6 months; this length was chosen as it reflects the average amount of time women stay in contact with DVA advocacy services. We have used a half-cycle correction.[15] A hypothetical cohort of 10 000 women aged 16 years or older was simulated moving between the states

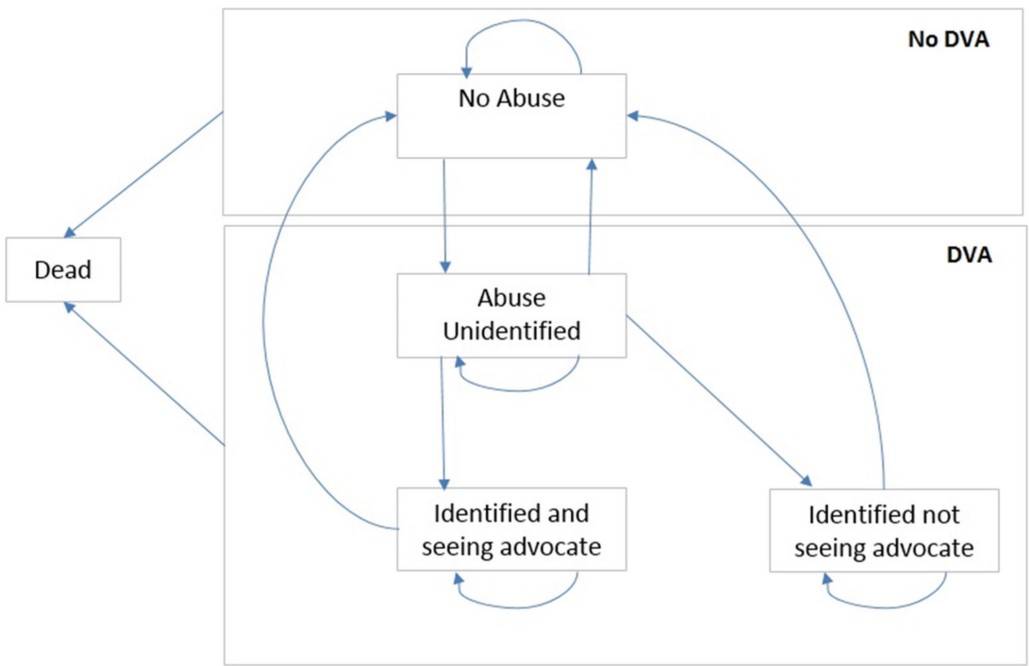

**Figure 1** Health states and movement between health states in Markov model. The model starts with all women in either the 'not abused' state or one of the states associated with abuse, based on the prevalence of domestic violence and abuse (DVA) (see text). Women in the 'not abused' state could stay in this state, move to 'abused but not identified' or die from any cause. Once women were in the 'abused but not unidentified' state, they could stay in that state, move back to 'not abused', move to 'abused and identified, seeing advocate' or 'abused and identified, not seeing advocate' or die. Women in the 'abused and identified' states could stay in these states, move back to 'not abused' or die.

(figure 1). Other than death, which is an absorbing state, women can transition between each of the other states 'not abused', 'abused but not identified', 'abused and identified, seeing advocate educator', 'abuse and identified, not seeing advocate educator'. As the hypothetical cohort of women aged 16 years or older were considered eligible for the intervention, all results were reported as 'per woman aged 16 years or older registered to GP practice'.

## Intervention

The IRIS programme is a multicomponent intervention that has been described in detail elsewhere.[10 11] In brief, it consists of two 2-hour multidisciplinary training sessions, for the practice clinical team and 1-hour training for reception and ancillary staff. They are delivered jointly by an IRIS advocate educator from a local collaborating specialist DVA agency, alongside a clinician interested in DVA, the IRIS clinical lead. The advocate educator is central to the intervention, combining a training and support role to the practices with provision of advocacy to women referred. Other intervention components include a simple 4-question questionnaire, carried out by the healthcare practitioner, addressing different aspects of DVA (humiliation, afraid, raped and kicked), such as "within the last year, have you been afraid of your partner of ex-partner?", also known as the HARK template[16] in the electronic medical record triggered by entry of clinical problem codes (such as depression, anxiety, irritable bowel syndrome, pelvic pain and assault), an explicit referral pathway to a named IRIS advocate educator and publicity materials about DVA visible in practices. Patients referred to the advocate educator are usually seen at the referring general practice, enhancing safety and confidentiality.

## Data collection and ethics approval

Several different data sources were used in this study. Whenever possible, we have used observational data from the IRIS programme. These were collected by IRIS team members, liaising with advocacy agencies and local authorities. Given that we only use anonymised data, arising from the usual care of women, individual consent of women was not required. This research project was given exemption from NHS Research Ethics processes, as it was classified as service evaluation. When observational data were unavailable, we have chosen to use peer-reviewed published data that was relevant to general practice and the UK. Each relevant parameter and its source are described in detail below.

## Prevalence of domestic abuse

The proportion of women aged 16 years or older experiencing abuse was estimated based on published epidemiological data. This was taken from a cross-sectional study carried out by Richardson et al in east London,[17] which reported a prevalence of 0.17 or 17% in the population of women consulting a GP or practice nurse. This is an estimate of the prevalence of DVA in general practice, generalisable for England.

## Transition probabilities

There are eight transitions between states in the model. Transition probabilities were obtained using observational data from the IRIS programme, the MOthers' Advocates In the Community programme,[10 18] the Office for National Statistics[19 20] and Health & Social Care Information Centre[21] and a Cochrane review,[13] evaluating the reduction of any type of domestic abuse with any type of advocacy. Observational data were obtained from commissioned IRIS sites that have been running for 2 years or more, where there was at least one full-time equivalent advocate educator and 20 general practices trained. It included six clinical commissioning groups in northern England, south-west England and London. Given the inclusion criteria, the sites represent the implementation of the programme. Table 1 provides the parameter values and their respective sources. Where no data were available, we have calculated estimates using the model calibration method described below.

## Model calibration

Because of uncertainty surrounding transition probabilities from *not abused* to *abused but not identified* and *vice versa*, we used the prevalence of abuse (17%) estimated in the study by Richardson et al,[17] to calibrate the model. The model was run for 3000 cycles, assuming that thereafter the number of women in each state would remain constant. This was based on our calculation of steady states. The transition probabilities from *not abused* to *abused but not identified* and *vice versa* were changed until the proportion of women in the *not abused* state exactly reflected the observed prevalence (100–17=83%). The initial distribution of women in the three *abused* states was also determined by this process.

## Utilities

Each state in the Markov model was associated with a utility score, which consisted of a general measure of health-related quality of life,[22] allowing us to measure QALYs associated with IRIS and the comparator based on the proportion of women in each health state in each of the 20 six-monthly cycles in the model, totalling 10 years. The utility score of women who were not abused was assumed to be 0.85.[23] Wittenberg et al conducted a cross-sectional survey to estimate community preferences for health states resulting from intimate partner violence. Using a UK-based algorithm, they found the utility of women experiencing any abuse was 0.64. When the severity/frequency of violence was low, the mean utility was 0.65 and when the severity/frequency was moderate or severe the mean utility was 0.63. For women who were abused in our model, we assumed this was moderate to severe, giving a utility score of 0.63.[24] For women seeing an advocate educator, we used the utility value of women with low abuse (0.65), implying that seeing an advocate

**Table 1** Model input parameters: probabilities, utilities and costs

| Parameter | Base case value | Lower limit | Upper limit | Distribution | Source | IRIS trial base value* |
|---|---|---|---|---|---|---|
| **Probabilities** | | | | | | |
| Proportion of women experiencing abuse | 0.17 | 0.147 | 0.194 | Beta | 17 | 0.17 |
| **Starting distribution for women who are abused** | | | | | | |
| Abused and identified, seeing advocate educator | 0.003† | 0 | 0.0066 | Uniform | ‡ | – |
| Abused and identified, not seeing advocate educator | 0.033† | 0 | 0.0660 | Uniform | ‡ | – |
| Abused but not identified | 0.964† | – | – | Uniform | Complement | – |
| **Transition probabilities** | | | | | | |
| Not abused to abused but not identified | 0.0037† | 0.0004 | 0.0106 | Dirichlet | ‡ | 0.0075 |
| Not abused to dead | 0.00551† | 0.0010 | 0.0136 | Dirichlet | 13 15 | 0.0058 |
| Stay in Not abused | 0.9908† | - | - | Dirichlet | Complement | 0.9867 |
| Abused but not identified to not abused (control) | 0.0500† | 0.0450 | 0.0553 | Dirichlet | ‡ | 0.025 |
| Abused but not identified to abused and identified, not seeing advocate educator (control) | 0.0027† | 0.0016 | 0.0040 | Dirichlet | IRIS-programme local sites | 0.0094 |
| Abused but not identified to abused and identified, seeing advocate educator (control) | 0.0005† | 0.0001 | 0.0011 | Dirichlet | IRIS-programme local sites | 0.0016 |
| Abused but not identified to dead (control) | 0.00554† | 0.0039 | 0.0074 | Dirichlet | 13 15 | 0.0059 |
| Stay in abused but not identified (control) | 0.9444† | - | - | Dirichlet | Complement | 0.9581 |
| Abused but not identified to not abused (intervention) | 0.0500† | 0.0450 | 0.0553 | Dirichlet | ‡ | 0.025 |
| Abused but not identified to abused and identified, not seeing advocate educator (intervention) | 0.0109† | 0.0086 | 0.0135 | Dirichlet | IRIS-programme local sites | 0.0207 |
| Abused but not identified to abused and identified, seeing advocate educator (intervention) | 0.0056† | 0.0040 | 0.0076 | Dirichlet | IRIS-programme local sites | 0.0101 |
| Abused but not identified to dead (intervention) | 0.00554† | 0.0039 | 0.0074 | Dirichlet | 6 | 0.0059 |
| Stay in abused but not identified (intervention) | 0.9419† | - | - | Dirichlet | Complement | 0.9383 |
| Abused and identified, seeing advocate educator to not abused | 0.1408† | 0.0707 | 0.2301 | Dirichlet | 18 | 0.0888 |
| Abused and identified, seeing advocate educator to dead | 0.00554† | 0.0000 | 0.0309 | Dirichlet | 13 15 | 0.0059 |
| Stay in abused and identified, seeing advocate educator | 0.8536† | - | - | Dirichlet | Complement | 0.9053 |
| Abused and identified, not seeing advocate educator to not abused | 0.0781† | 0.0136 | 0.1912 | Dirichlet | 18 | 0.0717 |
| Abused and identified, not seeing advocate educator to dead | 0.00554† | 0.0000 | 0.0438 | Dirichlet | 13 15 | 0.0059 |
| Stay in abused and identified, not seeing advocate educator | 0.9163† | - | - | Dirichlet | Complement | 0.9223 |
| **Utilities** | | | | | | |
| Not abused | 0.85 | 0.840 | 0.860 | Beta | 23 | – |
| Abused but not identified | 0.63 | 0.503 | 0.749 | Beta | 24 | – |
| Abused and identified, seeing advocate educator | 0.65 | 0.518 | 0.771 | Beta | 24 | – |

**Table 1** Continued

| Parameter | Base case value | Lower limit | Upper limit | Distribution | Source | IRIS trial base value* |
|---|---|---|---|---|---|---|
| Abused and identified, not seeing advocate educator | 0.63 | 0.503 | 0.749 | Beta | 24 | – |
| Costs | | | | | | |
| Costs of the intervention, per women registered, per 6 months | £0.46† | £0.01 | £1.69 | Gamma | IRIS-programme local sites | £0.55 |
| Cost of onward referral, once | £312† | £8 | £1127 | Gamma | IRIS-programme local sites & 11 | £298 |
| Cost of abused but not identified | £2043 | £52 | £7536 | Gamma | 8 | £4721 |
| Weighted costs abused and identified, seeing advocate educator | 1 | 0.75 | 1.25 | Gamma | Assumption | – |
| Weighted costs abused and identified, not seeing advocate educator | 1 | 0.9 | 1.1 | Gamma | Assumption | – |

Costs are in 2015/2016 UK£.
*Values obtained from Devine et al.[11]
†Value updated from Devine et al.[11]
‡Internal calculation based on model calibration.

educator slightly increased their quality-of-life scores. QALY gains were reported per woman aged 16 years or older registered to GP practice.

## Costs

We included intervention costs, costs of onward referral and costs associated with DVA (including costs to the UK NHS, lost economic output, costs to the criminal/civil justice system and personal costs). Costs were also reported per woman aged 16 years or older registered to GP practice.

One IRIS advocate educator typically provides training, support and advocacy services for 24 general practices at any one point in time. Intervention costs were calculated based on the actual budget of the IRIS programme in the six sites (including advocate educator salaries, travel, recruitment, laptop, telephone, publicity, clinician consultancy, evaluation and central management costs) at a total 6-month cost across all sites of £272 613. This was divided by the number of registered women aged 16+ years in IRIS-trained general practices in these sites (n=595 902). Costs of onward referral from the advocate educator was based on the finding of contact time from the IRIS trial, in which an onward referral was given to 57% of women in contact with an advocate educator and 63% of these women accepted this referral. Therefore, although costs of onward referral were based on current budgets and salaries, the proportion of contact was obtained from the trial estimates. Total costs per onward referral were therefore £861. Taking into account the proportion of women given a referral and accepting it,

and inflating it to 2015/2016 UK£, average costs of advocate educator contact per abused woman were £312.

Costs associated with intimate partner violence in the UK are described by Walby and Olive.[8] In their report, costs of lost economic output, health services, criminal justice system, civil justice system, social welfare, personal costs, specialised services and physical/emotional impact were individually reported, and total costs were €13 732 million (£11 billion) in 2012. We excluded costs of physical/emotional impact (€6614 million), as they were not financial costs, but consisted of monetary valuing of health status, which in cost-effectiveness models ought to be captured in terms of QALYs; these were also not included in the original cost-effectiveness analysis. The remaining costs were converted to UK£ and inflated to 2015/2016. Total costs per 6 months were £2933 million. Based on the 2015 Crime Survey for England and Wales, it was estimated that 1.3 million women experienced intimate partner violence in 2015/2016 in the UK.[3] Mean costs per abused woman were therefore £2043. We assumed that the costs of intimate partner abuse are similar to the costs of abuse by other family members, and that the costs would not differ between identified or unidentified abuse. In sensitivity analyses, we have allowed the costs of identified abuse to increase or decrease by 10% compared with abuse that was not identified; similarly, the costs of *abused and identified, seeing advocate educator* were allowed to increase or decrease by 25%.

## Cost-utility analysis

Costs and utilities were applied to each health state. Total costs and QALYs for the hypothetical cohort were

generated for the IRIS programme and the control group. The main outcome was the incremental costs per QALY gained. In the UK, an intervention is generally considered cost-effective when the incremental costs per QALY gained are <£20 000.[14] We also presented the results of cost-effectiveness analysis in terms of incremental net monetary benefit (NMB). This was calculated as the mean incremental QALYs per woman registered at the general practice accruing to IRIS multiplied by the decision-makers' maximum willingness to pay for a QALY (assumed to be £20 000), minus the mean incremental cost per woman. Negative incremental NMBs indicate that usual care was preferred on cost-effectiveness grounds and positive incremental NMBs favour IRIS.

The cost-utility analysis was conducted using pooled national data, but we have also evaluated the cost-effectiveness at different local sites. We allowed all parameters, including costs and benefits, to vary across sites and reported them individually.

### Sensitivity analysis

All parameters were varied in a one-way sensitivity analysis, using lower and upper limits based on 95% uncertainty intervals. We undertook a probabilistic sensitivity analysis, drawing random samples from the probability distributions of all parameters in 1000 simulations. All uncertainty intervals were calculated based on the 2.5th and 97.5th percentiles of the distribution of all the 1000 values in the probabilistic sensitivity analysis. The interpretation of these is different to that of statistical analysis CIs of clinical effects. In cost-effectiveness analysis, if an incremental cost-effectiveness ratio (ICER) has an uncertainty interval that crosses zero, it effectively means that the intervention can be cost-saving (negative value), cost-neutral (zero) or costly (positive value) per QALY gained. The proportion of simulations with an incremental cost per QALY gained below the cost-effectiveness threshold was calculated for different values, ranging from £0 to £50 000. The results were presented in a cost-effectiveness acceptability curve.

### Patient and public involvement

We did not directly include patient and public involvement (PPI) in this study, but the data collected from local IRIS programmes were developed with PPI.

## RESULTS
### Base case

Parameter values used in the base case analysis are shown in table 1, which also includes the parameters used in the original trial to allow for a direct comparison. The main differences between the parameters for this study and the trial parameters lie in the transition probabilities relating to the health state of 'abuse but not identified' and its cost.

Over the 10-year time horizon, mean total costs per woman were £4416 in the intervention group, compared with £4430 in the control group (table 2(a)). The IRIS programme therefore saves £14 per woman aged 16 years and older registered to GP practices, from a societal perspective over 10 years. Total QALYs per woman were 0.001 higher in the intervention group (6.671) than in the control group (6.669). Because the intervention was associated with lower costs and greater effectiveness, the incremental cost per QALY gained was negative (ie, IRIS dominates current practice as it is both cost-saving and more effective than usual care) and the incremental NMB was positive (£42). The incremental NMB was also positive (£22) when using an NHS-only perspective (table 2(b)).

Table 2 also presents the results for each site. The table shows that IRIS dominated current practice, from a societal perspective, in sites 1, 2, 3 and 4, with an incremental NMB of £41, £89, £29 and £59, respectively. From an NHS perspective, only in site 1 did IRIS dominate current practice, although it was cost-effective, using the threshold advised by the National Institute for Health and Care Excellence (NICE) of £20 000 per QALY gained, in sites 2 (ICER £2585 per QALY gained), 3 (ICER £3055 per QALY gained) and 4 (ICER £8317 per QALY gained). IRIS was found to be cost-effective (ICER £5882 per QALY gained) and borderline cost-effective (ICER £21 229 per QALY gained) from a societal and NHS perspectives, respectively, in site 5, and it was not cost-effective from either perspective in site 6 (ICER £52 557 per QALY gained and ICER £64 427 per QALY gained, respectively).

### Sensitivity analyses

Across all sites combined, results were most sensitive to varying the transition probability from *abused but not identified* to *not abused*. When in the control arm this was varied from 0.049 to 0.051, the incremental NMB varied from £110 to –£26 (figure 2). When it was varied similarly in the intervention arm, the incremental NMB varied from –£25 to £109. Figure 2 shows the 12 parameters that when varied had the highest impact on the incremental NMB.

Incremental costs and QALYs varied widely in probabilistic sensitivity analyses. The 95% uncertainty interval for incremental costs was –£151 to £37, for incremental QALYs it was –0.005 to 0.006 and for the incremental NMB it was –£247 to £351. Figure 3A shows a scatter plot of the incremental costs and incremental QALYs from the 1000 simulations. The IRIS programme is cheaper and more effective than the absence of the programme (usual care), dominating current practice in 35% of the simulations and was dominated by the absence of the programme in 18% of the simulations. The IRIS programme was cost-effective in 61% of simulations when the cost-effectiveness threshold was £20 000 (figure 3B).

## DISCUSSION
### Summary

We found that the IRIS GP training and service programme is likely to be cost-effective and cost-saving in the UK compared with usual care. The QALY gains

**Table 2** Base case results

| National IRIS (pooled results) | (a) Societal perspective | | | (b) NHS-only perspective | | |
|---|---|---|---|---|---|---|
| | Costs | QALYs | Cost-effectiveness | Costs | QALYs | Cost-effectiveness |
| Intervention (IRIS programme) | £4416 | 6.671 | | £1238 | 6. 671 | |
| Control (no programme) | £4430 | 6.669 | | £1232 | 6. 669 | |
| Difference (intervention vs control) | −£14 | 0.001 | −ve (intervention dominates control) | £6 | 0.001 | £3913 per QALY gained |
| Incremental NMB* | | | £42 | | | £22 |
| **Local site 1** | | | | | | |
| Intervention (IRIS programme) | £4318 | 6.671 | | £1231 | 6.671 | |
| Control (no programme) | £4334 | 6.669 | | £1232 | 6.669 | |
| Difference (intervention vs control) | −£16 | 0.001 | −ve (intervention dominates control) | −£1 | 0.001 | −ve (intervention dominates control) |
| Incremental NMB* | | | £41 | | | £26 |
| **Local site 2** | | | | | | |
| Intervention (IRIS programme) | £4305 | 6.673 | | £1240 | 6.673 | |
| Control (no programme) | £4333 | 6.670 | | £1232 | 6.670 | |
| Difference (intervention vs control) | −£28 | 0.003 | −ve (intervention dominates control) | £8 | 0.003 | £2585 per QALY gained |
| Incremental NMB* | | | £89 | | | £54 |
| **Local site 3** | | | | | | |
| Intervention (IRIS programme) | £4325 | 6.671 | | £1235 | 6.671 | |
| Control (no programme) | £4334 | 6.670 | | £1232 | 6.670 | |
| Difference (intervention vs control) | −£9 | 0.001 | −ve (intervention dominates control) | £3 | 0.001 | £3055 per QALY gained |
| Incremental NMB* | | | £29 | | | £17 |
| **Local site 4** | | | | | | |
| Intervention (IRIS programme) | £4326 | 6.672 | | £1253 | 6.672 | |
| Control (no programme) | £4334 | 6.669 | | £1232 | 6.669 | |
| Difference (intervention vs control) | −£8 | 0.003 | −ve (intervention dominates control) | £21 | 0.003 | £8317 per QALY gained |
| Incremental NMB* | | | £59 | | | £30 |
| **Local site 5** | | | | | | |
| Intervention (IRIS programme) | £4337 | 6.670 | | £1244 | 6.670 | |
| Control (no programme) | £4332 | 6.669 | | £1232 | 6.669 | |
| Difference (intervention vs control) | £4 | 0.001 | £5882 per QALY gained | £12 | 0.001 | £21 229 per QALY gained |
| Incremental NMB* | | | £6 | | | £0 |
| **Local site 6** | | | | | | |
| Intervention (IRIS programme) | £4395 | 6.671 | | £1307 | 6.671 | |
| Control (no programme) | £4334 | 6.670 | | £1232 | 6.670 | |
| Difference (intervention vs control) | £61 | 0.001 | £52 557 per QALY gained | £75 | 0.001 | £64 427 per QALY gained |
| Incremental NMB* | | | −£38 | | | −£52 |

Costs are in 2015/2016 UK£. Numbers may not sum due to rounding.
*Measured at a willingness to pay for a QALY of £20 000.
Costs are in 2015/2016 UK£. Numbers may not sum due to rounding.
IRIS, Identification and Referral to Improve Safety; NMB, net monetary benefit; QALY, quality-adjusted life- year.

associated with IRIS, which are average values for all eligible women aged 16 years or over registered at a practice (and not, eg, those who have been abused), are small; these are balanced against an equally small incremental cost of the intervention. Interventions with small costs and small gains are not uncommon in public health: a

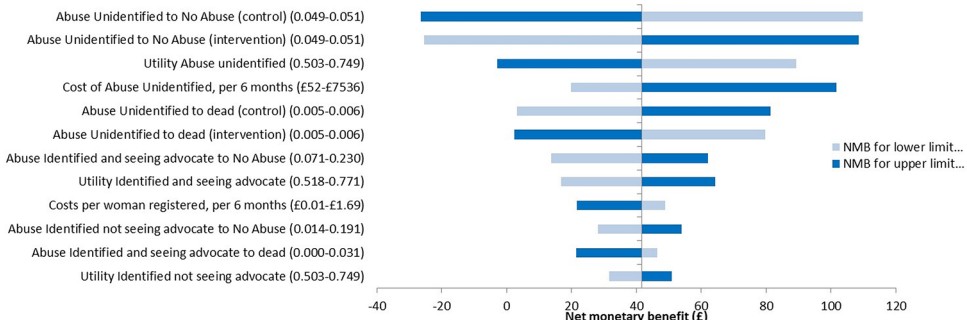

**Figure 2** Univariate sensitivity analysis. All analyses are as for the base case analysis with univariate adjustment of the parameters listed (see text). Results are point estimates of the incremental net monetary benefit (NMB) of the intervention vs control. The incremental NMB is calculated at a maximum willingness to pay for a quality-adjusted life-year of £20 000.

well-known example is influenza vaccination.[25 26] There is considerable uncertainty surrounding these results, but the probability that IRIS is cost-effective was >60% at the cost-effectiveness threshold commonly used in the UK. The cost-effectiveness acceptability curve is relatively flat, implying that the results from IRIS do not change much regardless of the threshold used. In our view, the shape of the cost-effectiveness analysis curve (CEAC) is entirely consistent with the 95% uncertainty intervals. The fact that these values are close to 50% reflects there is a high level of uncertainty, and the fact that the probability that IRIS is cost-effective is just >50% reflects the fact that IRIS is (slightly) favoured over the alternative according to our base case estimates. IRIS was more cost-effective when costs were measured from a societal perspective as the cost savings from reducing DVA were higher. IRIS was also cost-effective when taking an NHS-only perspective. There was some variation in value for money between sites, which appears to be driven mainly by the different rates of identification and/or referral, although different local costs have also contributed.

**Comparison with existing literature**

We contacted researchers in the field and searched the NHS Economic Evaluations Database and the HTA Database at the Centre for Reviews and Dissemination[27] for cost-effectiveness analyses of DVA programmes using the search terms 'domestic violence' and 'cost*' (28 August 2017). We identified four economic impact studies, all using modelling methods: one based on the pilot of the IRIS trial,[22] another based on the main trial,[11] the third based on an evaluation of independent domestic violence advisors[28] and the fourth of a trial of cognitive trauma therapy for abused women who have left the abusive relationship.[28] All the studies found the interventions cost-effective, despite uncertainty. Devine *et al* has reported a 75% probability of the DVA intervention being cost-effective,[11] while Mallender *et al* reported two scenarios out of possible five in which the intervention is not cost-effective.[28] Our findings are consistent with these previous studies. Our study is the only one that analyses the economic impact of a primary care-based programme implemented outside of trial settings.

**Strengths and limitations**

Our analysis has the strength of being based on a previously published cost-effectiveness model, updated with real-life data. Importantly, intervention costs and the probability of referral with IRIS were based on actual clinical practice, rather than in a research setting. We

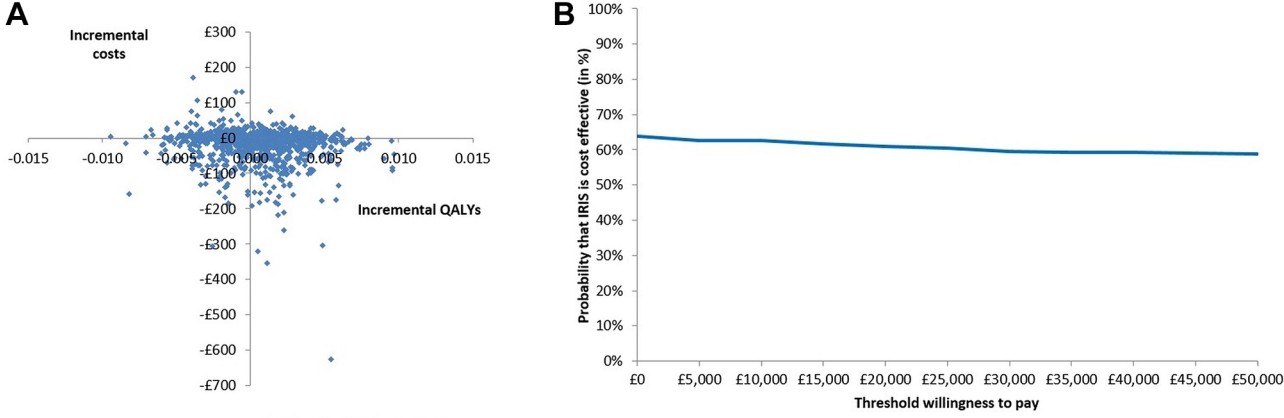

**Figure 3** Probabilistic sensitivity analysis. (A) Scatter plot of incremental costs and incremental quality-adjusted life-years (QALYs) from 1000 simulations. (B) Cost-effectiveness acceptability curve showing the probability in percentage terms that the intervention is cost-effective vs control at different values of the maximum willingness to pay for a QALY. Costs are in 2015/2016 UK£.

also had new data for the probability of identifying abuse and for what happened to women who were abused in current practice without the programme. However, it was not possible to update all parameter values. In particular, we were unable to update the utility value estimates, although in the sensitivity analysis, we have allowed these to vary and results were relatively stable. Costs of the intervention were calculated by dividing the total costs of the programme over all registered women in practices with the IRIS programme. Many of these women will never experience abuse and therefore cannot directly benefit from the programme. If programme costs were divided over women experiencing abuse only, mean costs per woman would be higher. However, the QALYs gained would also be higher, as these are also calculated for all women in the practices rather than just those who were abused. In fact, we have attempted to calculate these results dividing cost and QALYs over women experiencing abuse and the final ICER was unchanged, as both the numerator and denominator change by the same proportion. We did not include any impact of the IRIS programme on children exposed to DVA, as to our knowledge, there are no available cohort studies focusing on the costs and benefits of DVA interventions for this population, which might mean that we have underestimated the programme's cost-effectiveness. This was also highlighted in the NICE economic analysis of interventions to reduce incidence and harm of DVA: 'It can be expected there are likely to be additional benefits such as (to) the children and wider family members of victims of domestic violence' (p. 11).[28]

Another limitation is that we have used mainly data on short-term outcomes, although modelled long-term outcomes. There is unfortunately little data on long-term outcomes of DVA and the effect of advocacy, although it is generally agreed that effects last for a long time. This, however, bias our estimates against the intervention, implying our results are conservative.

### Implications for research and/or practice

The IRIS programme is likely to be cost-effective and cost-saving when implemented in the real life in the UK NHS. In order to decrease uncertainty around the cost-effectiveness estimates of IRIS and programmes like it, more data are needed on the utilities of women identified and women seeing an advocate and on long-term outcomes associated with DVA. Furthermore, future research should endeavour to understand the impacts and economic burden of DVA on exposed children, other family members and friends as well as focus on collecting up-to-date utility values for women subject to DVA in each health state.

Finally, our study has shown that there is moderate variation in the value for money of IRIS across different sites, implying qualitative research could focus on identifying the causes of such variation, in order to reduce it.

### Author affiliations
[1]University College London, Department of Applied Health Research, London, UK
[2]Division of Psychiatry, University College London, London, UK
[3]IRISi Interventions, Chepstow, UK
[4]NIHR CLAHRC North Thames at Bart's Health NHS Trust, Centre for Primary Care and Public Health, Queen Mary University of London, London, UK
[5]Pankhurst Trust Incorporating, Manchester Women's Aid, Manchester, UK
[6]Centre for Primary Care and Public Health, Queen Mary University of London, London, UK
[7]Centre for Tropical Medicine and Global Health, Nuffield Department of Clinical Medicine, University of Oxford, Oxford, UK
[8]Mahidol-Oxford Tropical Medicine Research Unit, Mahidol University, Bangkok, Thailand
[9]University of Exeter Medical School, Exeter, Devon, UK
[10]School of Social and Community Medicine, University of Bristol, Bristol, UK

**Acknowledgements** GF acknowledges the support of the National Institute for Health Research from a programme grant for applied research RP-PG-0614-20012 REPROVIDE (Reaching Everyone Programme of Research on Violence in diverse Domestic Environments). The authors would like to thank the IRIS partners who deliver the programme in the sites, especially those in northern England, south-west England and London who took the time and effort to provide with data.

**Contributors** SM, CG, SE, AS and GF have designed the study. EB, TV, SM, FS and AD have developed the Markov model and carried out the analysis of data. AS, FES, SD, CR, NVL and MJ have collected and validated the data. EB and SM have produced the initial draft. All authors have critically revised the manuscript and approved the final version.

**Funding** This research was funded by the National Institute for Health Research (NIHR) Collaboration for Leadership in Applied Health Research and Care North Thames at Barts Health NHS Trust.

**Competing interests** MJ has been paid by the IRIS project since 2007 for employment as an IRIS Advocate Educator and then as a National Implementation Manager. She is currently paid by IRISi, a social enterprise that is promoting the commissioning of the IRIS programme, for employment as Chief Executive. GF reports grants from National Institute for Health Research (NIHR), during the conduct of the study; and he is a non-executive board member of IRISi.

**Patient consent** Not required.

**Provenance and peer review** Not commissioned; externally peer reviewed.

**Data sharing statement** The anonymised data used in this study can be obtained from the corresponding author.

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
