## [Reviewer comments · BMJ Open]

ARTICLE DETAILS

TITLE (PROVISIONAL)	Cost-effectiveness of a domestic violence and abuse training and support programme in primary care in the real world: updated modelling based on a MRC phase IV observational pragmatic implementation study
AUTHORS	Capelas Barbosa, Estela; Verhoef, Talitha; Morris, Stephen; Solmi, Francesca; Johnson, Medina; Sohal, Alex; El-Shogri, Farah; Dowrick, Susanna; Ronalds, Clare; Griffiths, Chris; Eldridge, Sandra; Lewis, Natalia; Devine, A; Spencer, Anne; Feder, Gene

VERSION 1 – REVIEW

REVIEWER	N. Zoe Hilton Waypoint Centre for Mental Health Care, Canada
REVIEW RETURNED	22-Jan-2018

GENERAL COMMENTS	Overall  1. The large data base is a strength of this study. However, the overall results appear to be not statistically significant, and the QALY gains appear small. Currently the manuscript treats the project as effective, but notes almost as an afterthought that the confidence intervals are wide. In particular the sentence on p 16 “The IRIS programme is likely to be cost-effective” seems misleading given that the 95% CI indicates likelihood of a null effect. I think it is important to report these results regardless, but the manuscript should acknowledge the small effects, and discuss why that might be and what might need to change to improve the results. Abstract  2. Spell out IRIS and DVA the first time they are mentioned (this also applies in the introduction). 3. State briefly what the Markov model is (and provide a fuller explanation in the method section). Introduction  4. Is nonviolent abuse included in DVA? Do the sources of information about the health problems and costs of DVA include nonviolent abuse? Methods  5. Please state at the beginning of the method section, what the source of data was, who created the data and what the level of research ethics approval was. 6. What is a HARK template? 7. Multiple sources of data are introduced on pp 8-9 and these should be explained in more detail when they are first mentioned – which data were extracted from which sources? Do you have equal confidence in the reliability and relevance of each data source? Results  8. How does a QALY of one 100th of a year compare with other healthcare interventions? Does it translate into meaningful change in
---

	DV victims' lives? Discussion 9. The 60% probability of IRIS cost effectiveness seems difficult to reconcile with the 95% CI including zero, and with the essentially flat lines in Figure 3. Some further explanation may be required. 10. The results are compared with several comparison studies. Are all the cited studies based on data that are independent of the current study? Please provide more information about details to explain the sentence "All the studies found the interventions cost-effective, despite uncertainty." 11. Please explain the sentence "we have used mainly data on short-term outcomes, although modelled long-term outcomes." How will this affect the results and how does it limit the interpretation or generalization of results? Figures 12. Figures are small and difficult to use. Figure 1, please label the solid and dashed lines or explain in the figure note. Figure 3b, the Y axis is labelled "probability" but the axis markers are in percentages.
--	---

REVIEWER	Parveen Ali University of Sheffield, UK
REVIEW RETURNED	26-Feb-2018

GENERAL COMMENTS	In the methods section, the authors should mention QALY but for whom? I understand it refers to a used women but please clear state that. I am not confident of my interpretation of cost utility analysis and Markov modeling, however, the analysis makes sense.
--

REVIEWER	Steven McPhail Queensland University of Technology and Metro South Health, Australia
REVIEW RETURNED	04-Apr-2018

GENERAL COMMENTS	I have been requested to focus my review on the statistical / economic analyses reported in this paper. Summary: This manuscript represents a useful incremental advancement on the analyses reported by members of this authorship team in a prior modelling study. The prior study was primarily informed by a clinical trial. The present study advanced the prior Markov modelling by also including data from some sites that have now implemented the intervention in 'real world' clinical practice contexts. The authors are to be commended for taking the time to update their previous model using this latest health service data that was available to them. Overall, there seems to be a great deal of uncertainty, and the authors have discussed this clearly and openly, which is commendable. The analytical approaches seem generally appropriate in light of available data etc. On a small note before I move onto analysis specific comments, I just wanted to prompt the authors on two matters:  1. Was ethical approval required / obtained for this study (e.g., to access clinical data from participating sites where the intervention was implemented)? 2. I am not sure if the authors have completed a CHEERS checklist for uploading as supplementary material (I can't see it at the present
---

	time), but this may be worthwhile (depending on editorial policies and preferences which I am happy to defer to). Specific comments: 3. I am interested to know more about how much the 'real world' data updates (inputs) in the present study differed from the values used in their prior study (In this comment I am referring to inputs rather than the overarching conclusions). Were your prior estimates similar / dissimilar to the real-world data? 4. I suspect the study outcome hinges on some of the transition probabilities and utility / QALY estimates related to the health state of being abused but not identified +/- abused and seeing an advocate. The authors have rightly highlighted uncertainty around these issues and the absence of robust prior research from which estimates for the modelling could be drawn for some parameters. I wonder if it would assist the readership of BMJ Open (who are not primarily health economists) if the authors were able to highlight any precedents or recommendations from prior research regarding the general approach the authors have adopted for defining model parameters for which there is still no prior research was available to inform their modelling. 5. Were there any adjustments (e.g., a half-cycle correction) applied (or that should have been applied) that have not yet been described? 6. In addition to aforementioned uncertainty related to parameter inputs (and outputs), it seems there is also genuine heterogeneity in cost-effectiveness across sites. I would be interested to read a bit more about what the authors consider likely to be driving that. 7. If the authors had space within the manuscript text (e.g., discussion), it may be worthwhile to give their perspective on priorities for future research that may help to reduce key elements of uncertainty that have potential to sway the study findings (e.g., utility estimates).
--	---

VERSION 1 – AUTHOR RESPONSE

Reviewers' Comments to Author:

Reviewer: 1

Reviewer Name: N. Zoe Hilton

Institution and Country: Waypoint Centre for Mental Health Care, Canada

Competing Interests: None declared

Overall

1. The large data base is a strength of this study. However, the overall results appear to be not statistically significant, and the QALY gains appear small. Currently the manuscript treats the project as effective, but notes almost as an afterthought that the confidence intervals are wide. In particular the sentence on p 16 “The IRIS programme is likely to be cost-effective” seems misleading given that the 95% CI indicates likelihood of a null effect. I think it is important to report these results regardless, but the manuscript should acknowledge the small effects, and discuss why that might be and what might need to change to improve the results.

Thank you for your comment. There are 2 issues in the comment raised.

The first one refers to the small magnitude of QALY gains. We now acknowledge this explicitly in the first paragraph of the discussion, explaining that these are the average values across all women aged 16 or over registered to GP practices; not for example only in women who have been abused (we also make this clearer on pp7-8). We also explain that these small benefits are balanced against an equally small incremental cost.

The second issue raised refers to the uncertainty intervals. Unlike in statistical analysis, in cost-effectiveness analysis, these intervals are calculated from the probabilistic sensitivity analysis, and the interpretation of these is different to that of a statistical analysis of clinical effects, where the results would indeed be interpreted to mean a non-significant difference. In cost-effectiveness analysis, uncertainty intervals reflect the type of variation that is being captured: it is not sampling variation, e.g., variation across patients in a trial, but variation across several model parameter values from a range of different sources at once. Also, the intervals we report are the 2.5th and 97.5th percentiles of the distribution of all the 1000 values in the probabilistic sensitivity analysis, and so are not computed in the usual way to a confidence interval. For example, in cost-effectiveness analysis, if an ICER has an uncertainty interval that crosses zero, it effectively means that the intervention can be cost-saving (negative value), cost-neutral (zero) or costly (positive value) per QALY gained. This has been explored in the literature (1-3) and our results are consistent with other evaluations. We have clarified this and amended the manuscript on page 13.

Abstract

2. Spell out IRIS and DVA the first time they are mentioned (this also applies in the introduction).

Thank you. We have amended this (page 4).

3. State briefly what the Markov model is (and provide a fuller explanation in the method section).

Thank you. We have amended this (page 4).

Introduction

4. Is nonviolent abuse included in DVA? Do the sources of information about the health problems and costs of DVA include nonviolent abuse?

Thank you for your comment. Yes, we include non-violent abuse in our analysis. The definition of DVA for this study is consistent with the Declaration on the Elimination of Violence against Women (4) which defines violence against women as: "Any act of gender-based violence that results in, or is likely to result in, physical or psychological harm or suffering to women, including threats of such acts, coercion or arbitrary deprivation of liberty, whether occurring in public or in private life" (Article 1). We have included a reference to this on page 6.

Methods

5. Please state at the beginning of the method section, what the source of data was, who created the data and what the level of research ethics approval was.

Thank you for your suggestion. We have included a data collection and ethics approval subheading on page 8.

6. What is a HARK template?

The HARK template is a simple questionnaire composed of only four questions, one addressing for each of the following aspects of DVA: Humiliation, Afraid, Raped and Kicked (HARK). It was described in detail elsewhere in the literature (5). We have modified the main text to clarify this is a simple questionnaire (page 8).

7. Multiple sources of data are introduced on pp 8-9 and these should be explained in more detail when they are first mentioned – which data were extracted from which sources? Do you have equal confidence in the reliability and relevance of each data source?

Thank you for your comment. Different sources of data were used for different parameters. For the prevalence of DVA and utility scores, we have used peer-review published information. For the transition probabilities, which reflect the effectiveness of the intervention compared to control, we used observational data obtained by local sites. The costs of the intervention were also obtained with local sites and the cost of abuse was obtained from a peer-reviewed published study. These are all described in pages 9-12. Table 1 also includes the data source for each parameter for clarity. We have confidence that the data sources used are the most appropriate and up-to-date available, and have investigated the impact of uncertainty in the parameter values used in sensitivity analysis.

Results

8. How does a QALY of one 100th of a year compare with other healthcare interventions? Does it translate into meaningful change in DV victims' lives?

As mentioned previously, the small magnitude of the QALY gains directly reflect the explicit choice of using the total of eligible women in the cohort as the denominator. From an economic evaluation point of view the important point is that the modest benefits per eligible woman ought to be balanced also against the equally modest costs, and we have clarified this in the Discussion. There are other examples of interventions with small health benefits that have been balanced against small costs and shown to be cost-effective: a well-known example is flu vaccination (6, 7).

Discussion

9. The 60% probability of IRIS cost effectiveness seems difficult to reconcile with the 95% CI including zero, and with the essentially flat lines in Figure 3. Some further explanation may be required.

In our view the shape of the CEAC is entirely consistent with the 95% uncertainty intervals. A 60% probability that IRIS is cost-effective means there is a 40% chance that it is not. The fact that these values are close to 50% reflects there is a high level of uncertainty, and the fact that the probability that IRIS is cost-effective is just higher than 50% reflects the fact that IRIS is (slightly) favoured over the alternative according to our base case estimates.

We have included a brief explanation in the main text (page 15).

10. The results are compared with several comparison studies. Are all the cited studies based on data that are independent of the current study? Please provide more information about details to explain the sentence "All the studies found the interventions cost-effective, despite uncertainty."

Thank you for your comment. All cited studies use data that are independent from the current study, although there may be some overlap when peer-reviewed published data was used. Similarly to ours, the four studies mentioned have also conducted sensitivity analysis in which they found there to be considerable variation. We now report results from the other studies to justify the above statement.

11. Please explain the sentence "we have used mainly data on short-term outcomes, although modelled long-term outcomes." How will this affect the results and how does it limit the interpretation or generalization of results?

Unfortunately, to our knowledge, there is no information on long-term utilities scores for women who have experienced DVA. Therefore, we had to speculate that short-term outcomes were good proxies and could be used to model long term effects. This, however, can potentially bias our estimation of QALY gains. In one hand, one could imagine that after a certain time, women who had experienced abuse have the same utility score as women who were not abused, as the effects of abuse may have washed off. On the other hand, as research suggest (8), the effects of abuse may be more profound and last much longer than the time women are actively being abused. We believe this is more likely to be the case, implying that our results underestimate the effects of IRIS, making them conservative results. We have clarified this on page 17. Note also that we have accounted for this in our sensitivity analyses.

Figures

12. Figures are small and difficult to use. Figure 1, please label the solid and dashed lines or explain in the figure note. Figure 3b, the Y axis is labelled "probability" but the axis markers are in percentages.

Thank you for your suggestion, we have amended the Figures.

Reviewer: 2

Reviewer Name: Parveen Ali

Institution and Country: University of Sheffield, UK

Competing Interests: None

In the methods section, the authors should mention QALY but for whom? I understand it refers to a used women but please clear state that.

Thank you for your comment. We have clarified this on page 10.

I am not confident of my interpretation of cost utility analysis and Markov modeling, however, the analysis makes sense.

Reviewer: 3

Reviewer Name: Steven McPhail

Institution and Country: Queensland University of Technology and Metro South Health, Australia

Competing Interests: None declared

I have been requested to focus my review on the statistical / economic analyses reported in this paper.

Summary: This manuscript represents a useful incremental advancement on the analyses reported by members of this authorship team in a prior modelling study. The prior study was primarily informed by a clinical trial. The present study advanced the prior Markov modelling by also including data from some sites that have now implemented the intervention in 'real world' clinical practice contexts. The authors are to be commended for taking the time to update their previous model using this latest health service data that was available to them. Overall, there seems to be a great deal of uncertainty, and the authors have discussed this clearly and openly, which is commendable. The analytical approaches seem generally appropriate in light of available data etc. On a small note before I move onto analysis specific comments, I just wanted to prompt the authors on two matters:

1. Was ethical approval required / obtained for this study (e.g., to access clinical data from participating sites where the intervention was implemented)?

Thank you for your comment. Given that the cost-effectiveness analysis only uses anonymised data, arising from the usual care of women, individual consent of women was not required. Furthermore, this research was given exemption from NHS Research Ethics processes, as it was classified as service evaluation. We have included this information on page 8.

2. I am not sure if the authors have completed a CHEERS checklist for uploading as supplementary material (I can't see it at the present time), but this may be worthwhile (depending on editorial policies and preferences which I am happy to defer to).

Thank for your suggestion. We have now included a CHEERS checklist in the supplementary materials.

Specific comments:

3. I am interested to know more about how much the 'real world' data updates (inputs) in the present

study differed from the values used in their prior study (In this comment I am referring to inputs rather than the overarching conclusions). Were your prior estimates similar / dissimilar to the real-world data?

Thank you for your comment. In table 1, we have also included the parameters used in the trial in order to allow for a direct comparison. We have also highlighted this in the main text on page 12.

4. I suspect the study outcome hinges on some of the transition probabilities and utility / QALY estimates related to the health state of being abused but not identified +/- abused and seeing an advocate. The authors have rightly highlighted uncertainty around these issues and the absence of robust prior research from which estimates for the modelling could be drawn for some parameters. I wonder if it would assist the readership of BMJ Open (who are not primarily health economists) if the authors were able to highlight any precedents or recommendations from prior research regarding the general approach the authors have adopted for defining model parameters for which there is still no prior research was available to inform their modelling.

Thank you for your suggestion. We have added a new subheading 'Data collection' where we explain our approach (pages 8 and 9).

5. Were there any adjustments (e.g., a half-cycle correction) applied (or that should have been applied) that have not yet been described?

Thank you for your comment. We had indeed used the half-cycle correction. We have highlighted this in the main text on page 7.

6. In addition to aforementioned uncertainty related to parameter inputs (and outputs), it seems there is also genuine heterogeneity in cost-effectiveness across sites. I would be interested to read a bit more about what the authors consider likely to be driving that.

Thank you for your comment. Regarding the variation in the results from local sites, it appears that most differences came from the rate of identification and/or referral and the local intervention costs, the rates being more relevant than the costs. However, we have not conducted qualitative analysis to understand the mechanism driving this variation. We have included a mention to the most important factors in the text (page 15).

7. If the authors had space within the manuscript text (e.g., discussion), it may be worthwhile to give their perspective on priorities for future research that may help to reduce key elements of uncertainty that have potential to sway the study findings (e.g., utility estimates).

Thank you for your suggestion. We have amended the text accordingly (page 17).

References

1. Fenwick E, Claxton K, Sculpher M. Representing uncertainty: the role of cost-effectiveness acceptability curves. *Health economics*. 2001;10(8):779-87.
2. Briggs AH, Gray AM. Methods in health service research: Handling uncertainty in economic evaluations of healthcare interventions. *BMJ: British Medical Journal*. 1999;319(7210):635.
3. Briggs A, Fenn P. Confidence intervals or surfaces? Uncertainty on the cost-effectiveness plane. *Health economics*. 1998;7(8):723-40.
4. Assembly UG. Declaration on the Elimination of Violence against Women. UN General Assembly. 1993.
5. Sohal H, Eldridge S, Feder G. The sensitivity and specificity of four questions (HARK) to identify intimate partner violence: a diagnostic accuracy study in general practice. *BMC family practice*. 2007;8(1):49.
6. Muennig PA, Khan K. Cost-effectiveness of vaccination versus treatment of influenza in healthy adolescents and adults. *Clinical infectious diseases*. 2001;33(11):1879-85.
7. Sander B, Gyldmark M, Aultman R, Aoki FY. Impact on health outcome and costs of influenza treatment with oseltamivir in elderly and high-risk patients. *Journal of Medical Economics*. 2004;7(1-

4):67-83.

8. Walby S, Olive P. Estimating the costs of gender-based violence in the European Union: European Institute for Gender Equality; 2014.

VERSION 2 – REVIEW

REVIEWER	N Zoe Hilton Waypoint Centre for Mental Health Care, Canada.
REVIEW RETURNED	11-Jun-2018

GENERAL COMMENTS	Thank you for your revisions to this manuscript. It took some time to locate the response to reviewer comments on the earlier manuscript, as for some reason this was with the original submission record not the revised submission record but revised it when I found the letter. Page references in the response letter do not concur with the revised manuscript (either the footer or the pdf-generated header numbers) which also contributed to some delay. 1. The explanation in the response letter about the basis and interpretation of results is helpful, although the interpretation of uncertainty intervals and confidence intervals including zero are similar. Some of this explanation should be included in the actual manuscript, for the many researchers and practitioners interested in domestic violence interventions who may be drawn to your report but not familiar with the analytical approach. For example, it is appreciated that you mention the small benefits are balanced against a small cost (p. 16/17) but it would be better to also include the explanation of why this is key, and relate it to other examples like flu vaccination in your actual discussion (or introduction) that you mention in your response letter. 2. It's helpful to have more detail about the definition of abuse. Please consider spelling it out in the introduction in addition to referring to another source. For example, you could say "domestic violence and abuse including non-physical forms of abuse..." 3. Inclusion of ethics and data source information is appreciated. 4. Thank you for spelling out HARK. Is this completed by the patient alone or with the practitioner? Including an example of the items would be helpful to the reader.
---

REVIEWER	Steven McPhail Queensland University of Technology and Metro South Health, Australia
REVIEW RETURNED	08-Jun-2018

GENERAL COMMENTS	The authors have addressed my comments on their earlier submission.
---

VERSION 2 – AUTHOR RESPONSE

Reviewers' Comments to Author:

Reviewer: 1

Reviewer Name: Zoe Hilton

Institution and Country: Waypoint Centre for Mental Health Care, Canada.

Competing Interests: None declared

1. The explanation in the response letter about the basis and interpretation of results is helpful, although the interpretation of uncertainty intervals and confidence intervals including zero are similar. Some of this explanation should be included in the actual manuscript, for the many researchers and practitioners interested in domestic violence interventions who may be drawn to your report but not familiar with the analytical approach. For example, it is appreciated that you mention the small benefits are balanced against a small cost (p. 16/17) but it would be better to also include the explanation of why this is key, and relate it to other examples like flu vaccination in your actual discussion (or introduction) that you mention in your response letter.

Thank you for your suggestion. We have included some more details in about the interpretation of uncertainty intervals (p.14) and the flu vaccination example (p. 16).

2. It's helpful to have more detail about the definition of abuse. Please consider spelling it out in the introduction in addition to referring to another source. For example, you could say "domestic violence and abuse including non-physical forms of abuse..."

Thank you for your suggestion. We have included more detail in the definition of abuse as suggested (p.7).

3. Inclusion of ethics and data source information is appreciated.

Thank you very much.

4. Thank you for spelling out HARK. Is this completed by the patient alone or with the practitioner? Including an example of the items would be helpful to the reader.

Thank you for your suggestion. The questionnaire is completed with a healthcare practitioner, often within a consultation. We have included an example as suggested (p. 9).

Reviewer: 3

Reviewer Name: Steven McPhail

Institution and Country: Queensland University of Technology and Metro South Health, Australia

Competing Interests: None declared

The authors have addressed my comments on their earlier submission.

Thank you.

VERSION 3 – REVIEW

REVIEWER	Zoe Hilton Waypoint Centre for Mental Health Care, Canada
REVIEW RETURNED	05-Jul-2018
GENERAL COMMENTS	Thank you for responding to my comments and questions in detail. I now think that a non-specialist would be able to understand the study fully and its contribution to the domestic violence field.